# A Lightweight Image Super-Resolution Reconstruction Algorithm Based on the Residual Feature Distillation Mechanism

**DOI:** 10.3390/s24041049

**Published:** 2024-02-06

**Authors:** Zihan Yu, Kai Xie, Chang Wen, Jianbiao He, Wei Zhang

**Affiliations:** 1School of Electronic Information and Electrical Engineering, Yangtze University, Jingzhou 434023, China; 202101350@yangtzeu.edu.cn; 2School of Computer Science, Yangtze University, Jingzhou 434023, China; 400100@yangtzeu.edu.cn; 3School of Computer Science, Central South University, Changsha 410083, China; jbhe@mail.csu.edu.cn; 4School of Electronic Information, Central South University, Changsha 410083, China; csuzwzbn@csu.edu.cn

**Keywords:** super-resolution, spatial attention, residual feature distillation, image processing, global fusion

## Abstract

In recent years, the development of image super-resolution (SR) has explored the capabilities of convolutional neural networks (CNNs). The current research tends to use deeper CNNs to improve performance. However, blindly increasing the depth of the network does not effectively enhance its performance. Moreover, as the network depth increases, more issues arise during the training process, requiring additional training techniques. In this paper, we propose a lightweight image super-resolution reconstruction algorithm (SISR-RFDM) based on the residual feature distillation mechanism (RFDM). Building upon residual blocks, we introduce spatial attention (SA) modules to provide more informative cues for recovering high-frequency details such as image edges and textures. Additionally, the output of each residual block is utilized as hierarchical features for global feature fusion (GFF), enhancing inter-layer information flow and feature reuse. Finally, all these features are fed into the reconstruction module to restore high-quality images. Experimental results demonstrate that our proposed algorithm outperforms other comparative algorithms in terms of both subjective visual effects and objective evaluation quality. The peak signal-to-noise ratio (PSNR) is improved by 0.23 dB, and the structural similarity index (SSIM) reaches 0.9607.

## 1. Introduction

Image super-resolution (SR) reconstruction refers to the process of recovering a high-resolution (HR) image with more high-frequency information from one or multiple degraded low-resolution (LR) images. As an important means to improve image resolution, it solves the problem of obtaining high-resolution images in practical situations due to insufficient performance of acquisition devices or interference from external environments. It has been widely applied in fields such as intelligent surveillance [1], medical imaging [2], and target tracking [3]. However, the hardware devices for image acquisition have limitations and are expensive [4]. In contrast, signal processing-based super-resolution reconstruction algorithms are more flexible and cost-effective. There are two main categories of image super-resolution reconstruction algorithms: single-image super-resolution (SISR) and multi-image super-resolution (MISR). This study focuses on SISR reconstruction algorithms. However, SISR is a highly ill-posed inverse problem with a non-unique solution space. This is because a significant amount of high-frequency information is lost during the down-sampling process from the original image to obtain the LR image, resulting in insufficient usable information for the recovery process.

To address this inverse problem, numerous super-resolution reconstruction methods have been proposed. Currently, SISR (single-image super-resolution) algorithms can be broadly classified into three categories: interpolation methods [5,6,7,8], reconstruction methods [9,10,11,12], and learning-based approaches [13,14]. Interpolation-based methods utilize surrounding pixel information to predict unknown pixels based on the assumption of image continuity. Although easy to implement, these methods have limited linear model fitting capabilities, often resulting in blurry edges, contours, and inadequate texture restoration. Reconstruction-based methods primarily constrain the reconstruction results using image prior information, improving the blurring effect. However, they introduce computational complexity and still provide suboptimal performance for complex-structured images. Learning-based methods learn the mapping relationship between high- and low-resolution images from samples. In recent years, deep learning-based approaches have demonstrated remarkable achievements in the field of image super-resolution.

Despite the significant success achieved by CNN-based methods, most of them are unsuitable for mobile devices. Furthermore, the majority of current algorithms blindly increase the depth of the network, resulting in an excessive number of parameters and increased training difficulty. With the popularity of mobile devices and the development of edge computing, there is an increasing demand for efficient computation and processing in resource-constrained environments. In this context, lightweight models can offer several important advantages: saving computational resources and energy consumption, accelerating inference speed, reducing model storage space, and providing real-time edge intelligence.

To meet the aforementioned requirements, we propose a lightweight single-image super-resolution reconstruction network based on the residual feature distillation mechanism, aiming to achieve superior SISR reconstruction results with minimal network parameters and computational burden. The network is primarily composed of a residual feature distillation block (RFDB). Within each RFDB, we design a novel feature distillation method, mainly implemented by the residual feature distillation layer. Additionally, local residual learning (LRL) is added to each residual block to facilitate capturing fine-grained feature changes. Finally, a customized spatial attention module (SA) is added to the end of the RFDB to provide more available information for recovering high-frequency details such as image edges and textures. After multiple rounds of residual feature distillation, global feature fusion (GFF) is performed to adaptively maintain hierarchical features at a global scale.

In summary, the contributions of this paper can be summarized as follows:We propose a single-image super-resolution network (SISR-RFDM) based on the residual feature distillation mechanism. It achieves fast and accurate image super-resolution, demonstrating competitive results with a moderate number of parameters in the SISR task.We design an attention module (SA) that focuses on spatial regions, treating areas containing abundant information such as boundaries and textures differently. This allows the network to concentrate more on these regions, providing more useful information for image detail recovery.We introduce the global feature fusion (GFF) structure, which globally fuses the output features of each residual block. Using hierarchical feature fusion, we reduce feature redundancy and enhance inter-layer information flow and feature reuse.

The remainder of this paper is organized as follows. Section 2 presents related work. Section 3 presents the details of each module used in the proposed model. The experiment results and analysis are discussed in Section 4, and conclusions are presented in Section 5.

## 2. Related Work

### 2.1. Single-Image Super-Resolution Based on Deep Learning

With the rapid development of deep learning, numerous methods based on convolutional neural networks (CNNs) have become mainstream in SISR. Dong et al. [15] first introduced the use of CNNs for image super-resolution reconstruction and proposed the Super-Resolution Convolutional Neural Network (SRCNN), which utilizes three convolutional layers to achieve a nonlinear mapping between LR and HR image pairs. However, the network is shallow, extracting only limited local features. Additionally, the entire reconstruction process is performed in the HR space, as the network is trained on LR images upsampled to the target size using bicubic interpolation [7]. This results in high computational complexity and a slow training speed. To address this issue, Dong et al. [16] proposed the Fast Super-Resolution Convolutional Neural Network (FSRCNN), which directly takes LR images as inputs and uses deconvolution layers for upsampling at the end of the network. This significantly reduces the computational complexity and accelerates the network training speed, with a reconstruction time of only 1/38 compared with SRCNN. Shi et al. [17] introduced an efficient Sub-Pixel Convolutional Neural Network (ESPCN), which first convolves the input features to expand the feature channels to obtain $r^{2}$ feature maps. These maps are then rearranged along the channel axis to obtain feature maps enlarged by a factor of r, greatly improving the reconstruction efficiency compared with deconvolution layers. As a result, many current algorithms use sub-pixel convolutional operations for upsampling.

As the network depth increases, the residual network (ResNet) proposed by He et al. [18] mitigates the problem of gradient vanishing or explosion caused by the increase in convolutional layers. Inspired by ResNet, Kim et al. [19] introduced a very deep super-resolution reconstruction algorithm called VDSR. By simply stacking 20 convolutional layers and using skip connections, the algorithm not only learns high-frequency residuals layer by layer but also accelerates network convergence, further improving the reconstruction performance. Subsequently, even deeper models emerged. For example, Lim et al. [20] proposed an enhanced deep super-resolution network (EDSR) that improved reconstruction performance by removing batch normalization layers in each residual block and adding a residual scaling layer to stabilize network training. However, excessively deep network layers result in a large number of parameters and make it difficult to extract deep features. To address this issue, Tai et al. [21] proposed a deep recursive residual network (DRRN) that achieves parameter sharing through recursive learning of multiple residual units, effectively controlling the number of network parameters. Nevertheless, increasing the number of network layers leads to feature redundancy. Tong et al. [22] proposed a super-resolution dense network (SRDenseNet) that alleviates feature redundancy by introducing dense skip connections that concatenate all layers in the network, enabling low-level and high-level feature reuse. Considering the interdependence and interaction of feature representations between different channels, Zhang et al. [23] proposed a very deep residual attention network (RCAN) that adaptively learns more useful channel features by introducing channel attention mechanisms. However, it does not take into account the differences in importance across different spatial positions. Features extracted by networks at different levels have information with varying receptive field sizes. To fully utilize these hierarchical features, Zhang et al. [24] proposed a dense residual network (RDN) that enhances information transmission between layers by fusing the input and output features of each layer within each residual dense block. However, this stacking-based local fusion method significantly increases computation. Li et al. [25] designed a multi-scale feature fusion network (MSRN) that extracts local features of different scales using convolutional kernels of different sizes. The reconstructed HR image obtained with global feature fusion contains more texture details but also slows down network operation.

In recent years, more and more research has focused on designing more efficient lightweight models. Kong et al. [26] introduced a classifier into the original SR model to classify the difficulty levels of restoring input image blocks into three categories: easy, medium, and difficult complexity levels, corresponding to different complexity SR networks. Song et al. [27] pioneered the use of additive networks for image super-resolution learning, avoiding a large number of multiplicative operations during the convolution process, thus significantly reducing floating-point computations. Hui et al. [28] proposed an Information Distillation Network (IDN), which captures features in the distillation block (DBlock), merges some features with the input features, and then passes them to the module’s tail through skip connections. Although this can reduce subsequent feature channels, network parameters, and computational complexity, the module only performs single-time information distillation and cannot accurately distinguish the features to be refined from the features that need to be transmitted across layers. Based on the IDN framework, the research team proposed a fast and lightweight Information Multi-distillation Network (IMDN) [29]. It gradually extracts hierarchical features within the Information Multi-distillation Block (IMDB) and aggregates them based on the importance of candidate features using an adaptive pruning method. Building upon this, Cheng et al. [30] introduced recursive cross-learning to enhance feature extraction, resulting in improved performance. Inspired by ordinary differential equations, He et al. [31] developed the OISR-RK2 network (ODE-inspired network design for single-image super-resolution) for SR reconstruction. In the DID structure (a nested Dense In Dense structure), Li et al. [32] proposed the fusion of feature information using nested dense structures. Gao et al. [33] combined convolutional neural networks with Transformer and presented the lightweight and efficient LB-Net (Lightweight Bimodal Network). Furthermore, Choi et al. [34], based on the Transformer architecture, used a sliding window technique to expand the receptive field, enabling the network to better restore degraded pixels. LatticeNet [35] adopted a reverse sequential connection strategy for feature fusion across different receptive fields. RFDN [36] applied residual feature distillation blocks, which are a variant of IMDB and are more powerful and flexible. DLSR [37] introduced a differentiable neural architecture search method to find more powerful fusion blocks based on RFDB.

It can be seen that feature fusion has played a crucial role in recent advancements. However, the aforementioned feature fusion strategies suffer from significant memory consumption since multiple relevant feature maps need to be stored in memory before aggregation. To accelerate inference speed and reduce memory consumption, we optimize our network backbone by designing a new residual feature distillation mechanism and enhance the feature representation of the model by incorporating spatial attention mechanisms.

### 2.2. Attention Mechanism

Attention mechanisms in deep networks originated from studies on human visual perception. Selective focus on specific portions of available information while disregarding others is referred to as attention in cognitive science. Attention mechanisms were initially applied in the visual domain in the 1990s and were subsequently reintroduced by Mnih et al. [38] in the field of deep learning. They have garnered increasing attention in computer vision in recent years. Human visual attention enables us to concentrate on regions with high resolution or discernibility in images, even when low-resolution backgrounds are present. Gradually, attention is dispersed across the entire image, enabling information inference. In computer vision, attention mechanisms are crucial for deep networks to learn the distribution patterns of key information, disregarding irrelevant details and focusing more on the inherent characteristics of the data.

Attention mechanisms can be categorized as strong attention mechanisms and soft attention mechanisms, with the latter being more commonly used. Soft attention mechanisms consist of two types. The first type is spatial attention, which focuses on different positions of the feature map with varying degrees of intensity. Mathematically, for a feature map of size H × W × C, spatial attention is represented by a h × w matrix, where each position’s value serves as a weight for the corresponding position in the original feature map. Multiplying element-wise yields the attention-enhanced feature map. The second type is channel attention, which operates primarily on channels. This attention mechanism assigns different levels of attention to various image channels. Mathematically, for a feature map of size H × W × C, channel attention is represented by a 1 × 1 × C matrix, with each position corresponding to a weight for the respective channel in the original feature map. Element-wise multiplication generates the attention-enhanced feature map. Low-frequency and high-frequency information are distributed differently across spatial locations in an image. Some regions, characterized by smooth textures, are relatively easy to restore, while others contain high-frequency details such as edges and textures, making restoration more challenging. Therefore, incorporating attention mechanisms in SISR enables the network to assign higher learning weights to regions containing high-frequency information during the learning process.

## 3. Methods

### 3.1. Network Overview

We propose an adaptive single-image super-resolution reconstruction algorithm, called SISR-RFDM (single-image super-resolution reconstruction algorithm based on the residual feature distillation mechanism), which uses an end-to-end approach to learn the mapping relationship between low-resolution and high-resolution images. The network consists of three main modules: a shallow feature extraction module, a deep feature extraction module, and a reconstruction module. The shallow feature extraction module uses a 3 × 3 convolutional layer to extract shallow features from the input low-resolution image. The deep feature extraction module consists of three cascaded RFDBs (residual feature distillation blocks). The concatenation of RFDB modules facilitates the extraction of deep hierarchical features. Within each RFDB, a multi-stage residual feature distillation mechanism is used to further extract deep features. To address information loss during the training of deep networks, the layered features outputted by the RFDBs are aggregated and dimensionally reduced using a 1 × 1 convolutional layer. Finally, global feature fusion (GFF) is applied to connect shallow and deep features to promote network convergence. Additionally, a spatial attention module is applied before obtaining these layered features to focus more on regions carrying high-frequency information. The reconstruction module comprises two 3 × 3 convolutional layers and a sub-pixel convolutional layer. At the end of the network, the sub-pixel convolutional layer is used for upsampling, enlarging the aggregated features to the target size, and improving the reconstruction efficiency of the model. The final output is the reconstructed image. The specific network architecture is illustrated in Figure 1.

### 3.2. Residual Feature Distillation Block

The designed RFDB is presented in Figure 2. To improve the quality of image reconstruction and make the model more lightweight and efficient, we introduce a series of lightweight and optimization strategies. Specifically, we incorporate the residual feature distillation mechanism and spatial attention module on top of the regular deep convolution.

In detail, we first perform channel separation on the input $F_{RFDB}^{i-1}$ and fuse all the distilled features to obtain $F_{distilled}$. Then, we calculate the spatial attention value $M_{SA}$ using the designed spatial attention module (SA) and weigh the different spatial positions of $F_{distilled}$ to obtain $F_{SA}$. This allows for better utilization of the spatial information of the input features, thereby further enhancing the accuracy and robustness of the model. Additionally, to smoothly propagate the features from the previous layer to the next layer, a short skip connection is introduced. Finally, the output $F_{RFDB}^{i}$ of the *i*-th RFDB is obtained using residual learning, further optimizing the performance of the model. This process can be represented as follows:(1)$$M_{SA}=SA\left(F_{distilled}\right)$$
(2)$$F_{SA}=F_{distilled}⊗M_{SA}$$
(3)$$F_{RFDB}^{i}=F_{SA}+F_{distilled}+F_{RFDB}^{i-1},i=1,2,3$$

By incorporating such lightweight and optimization strategies, we successfully reduced the complexity and parameter count of the model to a lower level while maintaining high reconstruction quality and performance. This not only enhances the versatility of the model but also makes it more suitable for a wide range of applications. Additionally, in resource-constrained scenarios, where computational resources are limited, these strategies allow for the model to be applied more effectively. The reduction in complexity and parameter count enables faster computations and the efficient utilization of available resources. Therefore, by introducing these lightweight and optimization strategies, we not only improved the model’s performance but also expanded its applicability in various practical settings.

#### 3.2.1. Residual Feature Distillation Mechanism

In the Information Multi-level Distillation Network (IMDN), we found that the information distillation operation is achieved using a 3 × 3 convolution, which compresses the feature channels at a fixed ratio. However, we discovered that using a 1 × 1 convolution to reduce the channels is more effective, as performed in many other CNN models. Inspired by the information distillation mechanism (IDM), in this section, we introduce a new residual feature distillation mechanism (RFDM).

As shown in Figure 2, we used a series of lightweight strategies to enhance the computational efficiency of the model while simultaneously reducing the parameter count. One of these strategies involves replacing the original 3 × 3 convolution operation on the left with 1 × 1 convolutions, which effectively compress feature channels during information distillation. This improvement significantly reduces the parameter count of the model while maintaining high reconstruction quality. The convolution on the far right still uses a 3 × 3 kernel, as it is located in the main body of the RFDB and needs to consider the spatial context for better feature refinement. Furthermore, we proposed a new residual feature distillation mechanism, utilizing two processing layers, namely, the distillation layer (DL) and the refinement layer (RL), to distill and refine input features, respectively. With this design, we can better utilize input features and further optimize the model’s performance. With the implementation of these lightweight strategies, we successfully improved the model’s computational efficiency and reduced its parameter count. This broadened the model’s potential applications and enhanced its performance, particularly in resource-constrained scenarios.

Specifically, we first use a 3 × 3 convolutional layer to extract the input features for subsequent distillation steps. For each distillation operation, we divide it into a distillation layer (DL) and a refinement layer (RL) to process the previous features. The DL is responsible for generating distilled features, while the RL further refines the features. This results in two parts of features, one preserved after the DL and the other sent to the next computational unit after the RL. Given the input feature $F_{RFDB}^{i-1}$, this process in the *i*-th RFDB can be described as follows: first, input feature $F_{RFDB}^{i-1}$ is passed through a 3 × 3 convolution layer to obtain $DL_{1}$ and $RL_{1}$, which yield the first-level distilled feature $F_{distilled\_1}$ and the feature to be refined, $F_{coarse\_1}$. Before $F_{coarse\_1}$ enters the next distillation unit, the feature to be refined undergoes channel expansion using a 1 × 1 convolution (to match the channel number of the input features) and further refinement of deep features using a residual block containing two convolutional layers. Finally, these refined features are separately passed through $DL_{2}$ and $RL_{2}$ to obtain the second-level distilled feature, $F_{distilled\_2}$, and the feature to be further refined, $F_{coarse\_2}$. Similarly, third-level distilled feature $F_{distilled\_3}$ and the feature to be refined $F_{coarse\_3}$ can be obtained. It is worth noting that a direct 3 × 3 convolution is applied to $F_{coarse\_3}$ to obtain the fourth-level distilled feature $F_{distilled\_4}$. This process can be expressed as
(4)$$F_{distilled\_1}=Conv^{1\times 1}\left\{LReLU\left[Conv^{3\times 3}\left(F_{RFDB}^{i-1}\right)\right]\right\}$$
(5)$$F_{coarse\_1}=LReLU\left[Conv^{3\times 3}\left(F_{RFDB}^{i-1}\right)\right]$$
(6)$$F_{distilled\_2}=Conv^{1\times 1}\left\{\begin{array}{l} Conv^{3\times 3}\left[LReLU\left\{Conv^{3\times 3}\left[Conv^{1\times 1}\left(F_{coarse\_1}\right)\right]\right\}\right]\\ +Conv^{1\times 1}\left(F_{coarse\_1}\right) \end{array}\right\}$$
(7)$$F_{coarse\_2}=\left\{\begin{array}{l} Conv^{3\times 3}\left[LReLU\left\{Conv^{3\times 3}\left[Conv^{1\times 1}\left(F_{coarse\_1}\right)\right]\right\}\right]\\ +Conv^{1\times 1}\left(F_{coarse\_1}\right) \end{array}\right\}$$
(8)$$F_{distilled\_3},F_{coarse\_3}=DL_{3}\left(F_{coarse\_2}\right),RL_{3}\left(F_{coarse\_2}\right)$$
(9)$$F_{distilled\_4}=Conv^{3\times 3}\left(F_{coarse\_3}\right)$$

In the above equation, $DL_{j}(•)$ represents the *j*-th layer of the distillation operation, $RL_{j}(•)$ represents the *j*-th layer of the refinement operation, $Conv^{1\times 1}(•)$ denotes the convolution operation with a 1 × 1 kernel, and $Conv^{3\times 3}(•)$ represents the convolution operation with a 3 × 3 kernel.

Finally, all distilled features $F_{distilled\_1}$, $F_{distilled\_2}$, $F_{distilled\_3}$, and $F_{distilled\_4}$ are fused along the channel dimension using a 1 × 1 convolution. This process can be described as follows
(10)$$F_{distilled}=\left\{\begin{array}{l} W_{distilled}^{1\times 1}\times Concat\left(F_{distilled\_1},F_{distilled\_2},F_{distilled\_3},F_{distilled\_4}\right)\\ +B_{distilled} \end{array}\right\}$$

#### 3.2.2. Spatial Attention Mechanism

The distribution of low-frequency and high-frequency information in various spatial positions of LR images does not align uniformly. Certain regions exhibit smoothness, making them comparatively easier to restore, while others entail numerous high-frequency details such as boundaries and textures, resulting in relatively challenging restoration. Hence, it becomes imperative to differentiate these regions and prioritize attention on areas carrying high-frequency information. Consequently, a spatial attention module, as illustrated in Figure 3, is devised to concentrate on specific spatial regions.

The spatial attention module first conducts average and standard deviation pooling separately on feature $F_{distilled}$ along the channel axis
(11)$$\bar{X}(i,j)=\frac{1}{C}\sum_{c=1}^{C}X_{c}(i,j)$$
(12)$$\sigma (i,j)=\sqrt{\frac{1}{C}\sum_{c=1}^{C}\left[X_{c}(i,j)-\bar{X}(i,j)\right]}$$

In the above equation, $\bar{X}(i,j)$ represents the result of channel-wise average pooling at spatial position (i,j); σ(i,j) represents the result of channel-wise standard deviation pooling at spatial position (i,j); $X_{c}(i,j)$ denotes the feature value at position (i,j) in channel c; and C represents the total number of channels. The two pooling results are then concatenated along the channel dimension, resulting in two sets of spatial feature descriptors, $F_{avg}$ and $F_{std}$. Next, a convolution layer (with a 5 × 5 kernel and stride of 1) is utilized to fuse the feature values at different positions within the feature descriptors and compress them into a single channel. Finally, the spatial attention map, $M_{SA}$, is obtained by applying the sigmoid activation function to normalize the output values between 0 and 1. These designs contribute to the light weight of the model. Specifically, applying pooling operations to the features reduces their dimensionality, thereby decreasing computational complexity. Moreover, the use of convolutional kernels for feature fusion helps prevent an excessive number of network parameters, further reducing the model size. Therefore, our SA module enhances both the reconstruction effectiveness and computational efficiency of the model while preserving its light-weight advantages. This process can be represented as follows
(13)$$M_{SA}=Sigmoid\left\{Conv^{1\times 1}\left\{LReLU\left\{Conv^{5\times 5}\left[Concat\left(F_{avg},F_{std}\right)\right]\right\}\right\}\right\}$$

In the above equation, Sigmoid(•) and LReLU(•) represent the activation functions of sigmoid and Leaky
ReLU, respectively. They are defined as
(14)$$Sigmoid(x)=\frac{1}{1+exp(-x)}$$
(15)$$LReLU(x)=\{\begin{array}{cc} x, & x\geq 0\\ alpha*x, & x<0,0<alpha<1 \end{array}$$

In the Leaky ReLU activation function, we set the initial slope alpha to 0.05.

### 3.3. Loss Function

To minimize the reconstruction error, we optimize the network using a loss function. There are various definitions of loss functions in the field of image super-resolution. We have considered the two most commonly used loss functions that are widely employed in most algorithms. The first one is a mean squared error (*MSE*), which is defined as
(16)$$l_{MSE}=\frac{1}{N}\sum_{i=1}^{N}{\left\|I_{i}-{\hat{I}}_{i}\right\|}_{2}^{2}$$

However, experiments conducted by Lim et al. [20] indicate that training with *MSE* loss is not a good choice because it penalizes large errors more and tolerates small errors better, resulting in over-smoothed images with a lack of high-frequency details. The second one is mean absolute error (*MAE*), defined as
(17)$$l_{MAE}=\frac{1}{N}\sum_{i=1}^{N}{\left\|I_{i}-{\hat{I}}_{i}\right\|}_{1}$$

Compared with *MSE* loss, *MAE* loss exhibits higher reconstruction performance and convergence. Therefore, we ultimately chose to optimize the model parameters using the *MAE* loss function. The optimization objective can be formulated as
(18)$$l_{MAE}\left(I_{HR},{\hat{I}}_{HR}\right)=\frac{1}{N}\sum_{i=1}^{N}{\left\|I_{HR}^{\left(i\right)}-{\hat{I}}_{HR}^{\left(i\right)}\right\|}_{1}$$
(19)$$\hat{\theta }=\underset{\theta }{argmin}\;\;l_{MAE}\left(I_{HR},{\hat{I}}_{HR}\right)$$

In the above equation, ${\hat{I}}_{HR}^{\left(i\right)}$ and $I_{HR}^{\left(i\right)}$ represent the reconstructed image and the corresponding ground-truth high-resolution image for the *i*-th sample, respectively; N represents the number of samples in the dataset; θ represents the parameters of the network that need to be learned; and $\hat{\theta }$ represents the parameters of the network after iterative updates.

## 4. Experimental Results and Analysis

### 4.1. Datasets and Metrics

Regarding the training process, there are various datasets available for single-image super-resolution. The most widely used ones are the 291-image set by Yang et al. [39] and the Berkeley Segmentation Dataset [40]. However, these datasets do not provide enough imagery to adequately train deep neural networks. Therefore, we opted to utilize the publicly available DIV2K dataset [41]. The DIV2K dataset consists of 800 training images, 100 validation images, and 100 testing images, all of which are high-quality. Due to its rich content, many SR models use DIV2K. For the testing phase, we evaluated our model’s performance on four widely used benchmark datasets: Set5 [42], Set14 [43], BSD100 [40], and Urban100 [44]. To assess the image reconstruction quality, we used the peak signal-to-noise ratio (*PSNR*) and the structural similarity index (*SSIM*) [45] as objective evaluation metrics. *PSNR* measures the pixel value error between the SR image and the corresponding HR image based on mean squared error (*MSE*). It is measured in decibels (dB) and defined as follows
(20)$$C_{PSNR}=10•lg\left(\frac{x_{max}^{2}}{E_{MAE}}\right)=20•lg\left(\frac{x_{max}}{\sqrt{E_{MAE}}}\right)$$
(21)$$E_{MAE}=\frac{1}{N}\sum_{i=1}^{N}{\left\|x_{i}-{\hat{x}}_{i}\right\|}^{2}$$

In the above equation, $x_{i}$ represents the pixel value at the *i*-th position in the HR image, ${\hat{x}}_{i}$ represents the pixel value at the *i*-th position in the SR image, N represents the total number of pixels in the image, and $x_{max}$ represents the maximum possible pixel value. The *PSNR* solely focuses on pixel differences without taking into account human visual perception. Therefore, we introduced the *SSIM* as a complementary evaluation metric. The *SSIM* quantifies the similarity between the SR image and the HR image, considering factors such as brightness, contrast, and structural information. It ranges from 0 to 1 and is defined as follows
(22)$$S_{SSIM}\left(x,\hat{x}\right)=\frac{\left(2\mu_{x}\mu_{\hat{x}}+c_{1}\right)\left(\sigma_{x\hat{x}}+c_{2}\right)}{\left(\mu_{x}^{2}+\mu_{\hat{x}}^{2}+c_{1}\right)\left(\sigma_{x}^{2}+\sigma_{\hat{x}}^{2}+c_{2}\right)}$$

In the above equation, x represents the HR image, $\hat{x}$ represents the SR image, $\mu_{x}$ and $\mu_{\hat{x}}$ denote the mean pixel values of the HR and SR images, $\sigma_{x}$ and $\sigma_{\hat{x}}$ represent the standard deviations of the pixel values in the HR and SR images, and $\sigma_{x\hat{x}}$ corresponds to the covariance between the HR and SR images. To maintain stability, we set constants $c_{1}={\left(k_{1}L\right)}^{2}$, $c_{2}={\left(k_{2}L\right)}^{2}$, and L to 255. Additionally, $k_{1}$ and $k_{2}$ are set to 0.01 and 0.03, respectively. The values of $C_{PSNR}$ and $S_{SSIM}$ are calculated in the Y channel of the YCbCr color space, which is derived from the RGB color space.

In addition to the *PSNR* and *SSIM*, we also introduced LPIPS (Learned Perceptual Image Patch Similarity) and FID (Fréchet Inception Distance) as additional evaluation metrics. LPIPS is a learned perceptual image patch similarity index that uses a pre-trained deep neural network to measure the perceptual difference between two images. A lower LPIPS value indicates a higher perceptual similarity between the SR (super-resolution) image and the HR (high-resolution) image. On the other hand, FID is a metric used to compare the similarity of two image distributions. It measures the difference between the feature distributions of generated and real images using a pre-trained inception network. A lower FID value indicates a higher distribution similarity between the SR and HR images.

By considering these evaluation metrics including the *PSNR*, *SSIM*, LPIPS, and FID, we can comprehensively evaluate the performance of super-resolution models in image reconstruction tasks.

### 4.2. Implementation Details

We randomly crop the LR images of size 48 × 48 from the DIV2K training set that have been interpolated by bicubic interpolation. To avoid overfitting, we perform data augmentation by randomly rotating the input image block by 90°, 180°, and 270°, as well as horizontally flipping it. During the training phase, we use the Adam algorithm [46] to update the model parameters with the following settings $\beta_{1}=0.9$, $\beta_{2}=0.999$, and $\varepsilon =10^{-8}$. We initialize a learning rate of $5\times 10^{-4}$ and train the model for 1000 epochs, halving the learning rate every 200 epochs. We set the model width to 64, and each batch is set to 8 inputs. The experimental conditions for our network include an 11th Gen Intel(R) Core(TM) i7-11800H @2.30GHz CPU (Santa Clara, CA, USA), an NVIDIA GeForce RTX 3060 GPU (Santa Clara, CA, USA), the Windows 10 operating system, and the PyTorch 2.0.1 deep learning framework.

### 4.3. Ablation Study

#### 4.3.1. Impact of the Residual Feature Distillation Module on the Network

To investigate the universal effectiveness of the RFDB module under different datasets and magnification conditions, experiments were conducted on the Set5 test set with a magnification factor of 2, the Set14 test set with a magnification factor of 3, and the BSD100 test set with a magnification factor of 4, while keeping other variables constant. The maximum PSNR values of the models without the RFDB module structure and the original network were compared, as shown in Figure 4.

The orange line in Figure 4 represents the PSNR value variation curve of the model using the RFDB module, while the blue-green line represents the PSNR value variation curve of the model without the RFDB module. Although the model without the RFDB module exhibited a fast convergence speed, it encountered overfitting as the training progressed, whereas the model using the RFDB module did not experience such a phenomenon. The experimental results indicate that compared with the network without the RFDB module, the network using the RFDB module demonstrated significant improvement in the maximum PSNR values under different test sets and magnification factors, leading to a noticeable enhancement in the overall performance of the network. Furthermore, the maximum SSIM values were also recorded simultaneously during the experiment for the three conditions, and the comparative results revealed an enhancement in SSIM values for the network using the RFDB module.

#### 4.3.2. Impact of Global Feature Fusion and Spatial Attention on the Network

To investigate the impact of the employed global feature fusion structure and spatial attention module on the final reconstruction results, relevant ablation experiments were conducted. Figure 5 illustrates the performance of the four models on the validation set during the training process. Here, “Base” refers to the base module, “GFF” represents the global feature fusion module, and “SA” represents the spatial attention module.

From the figure, it can be observed that with an increase in the number of training iterations, the corresponding PSNR (peak signal-to-noise ratio) values of the four models steadily improve. The final experimental results demonstrate that GFF and SA can further enhance the model’s performance, and coupling the use of SA and GFF maximizes the effectiveness of the model. It is worth mentioning that during the initial training stage, it is evident that the Base + GFF model achieves a leading advantage. This is mainly because it uses the global feature fusion structure, which allows for the rapid transmission of low-frequency features in the image to the network’s end, thereby expediting the reconstruction process.

Subsequently, the four obtained models were quantitatively analyzed on the Set5, Set14, BSD100, and Urban100 test sets, as shown in Table 1.

The results demonstrate that the inclusion of either GFF or SA on the Base module leads to improvements in the PSNR values. However, the network achieves the best performance when both modules are used together, exhibiting the most significant increase in the PSNR value of approximately 0.08 dB. These quantitative analyses provide evidence for the effectiveness of incorporating the global feature fusion module and the spatial attention module.

### 4.4. Comparison with State-of-the-Art Methods

To evaluate the image reconstruction effectiveness of our algorithm in this study, we compared it with other lightweight super-resolution (SR) methods, including Bicubic, SRCNN [15], FSRCNN [16], ESPCN [18], VDSR [20], DRRN [22], IMDN [29], RFDN [36], LBNet [33], NG-swin [34], and SwinIR-light [47]. Among them, the results of SRCNN, FSRCNN, ESPCN, VDSR, and DRRN were obtained by retraining using the same training data and techniques as our study. The results of IMDN, RFDN, LBNet, NG-swin, and SwinIR-light were obtained by testing with pre-trained models provided officially.

#### 4.4.1. Objective Quantitative Analysis

Table 2 presents the PSNR and SSIM values of the reconstructed images using our proposed algorithm and the comparative algorithms on four benchmark datasets for ×2, ×3, and ×4 upscaling factors. Higher PSNR and SSIM values indicate better reconstruction performance. The red text represents the best performance, while the blue text represents the second-best performance.

By comparing the data in the table, it can be observed that our proposed algorithm outperforms the other methods for most of the datasets, especially for scaling factors of ×3 and ×4. The PSNR values improved up to 0.14 dB compared with NG-swin, and the highest SSIM value reached 0.9613. These experimental results demonstrate the significant advantage and improved reconstruction performance of our algorithm in image super-resolution.

#### 4.4.2. Comparison of Additional Performance Metrics

To further verify the effectiveness and superiority of our algorithm, we introduced additional evaluation metrics LPIPS and FID. The LPIPS metric describes the perceptual similarity between SR images and HR images, while the FID metric considers the global feature distribution of the images. For different image distributions, smaller values of these metrics indicate that the generated image distribution is closer to the real image distribution, implying better image reconstruction quality. Table 3 presents the results of our evaluation using these metrics on the test set, along with the parameter count and average inference time for each algorithm, facilitating a comprehensive comparison.

From Table 3, we can see that our algorithm achieved outstanding results in all metrics, demonstrating its effectiveness and superiority in the super-resolution task. Particularly noteworthy is that our algorithm still achieves the best performance, even with a relatively small parameter count.

#### 4.4.3. Subjective Visual Perception

Figure 6, Figure 7 and Figure 8, respectively, illustrate the results of various algorithms (SRCNN, ESPCN, VDSR, IMDN, RFDN, SISR-RFDM) for ×2, ×3, and ×4 image super-resolution (SR) reconstruction. Additionally, ground-truth high-resolution (HR) images are provided for reference. To enable a more explicit comparison, we locally magnified the contents within the rectangular boxes.

From the figures, it is observable that the HR images obtained with Bicubic interpolation appear blurry with poor visual quality. Compared with algorithms such as RFDN, there is more severe distortion in the local details of the reconstructed images. In contrast, our proposed algorithm effectively restores fine details such as edges and textures. For instance, as demonstrated in the reconstructed comparison figure within the rectangular box for a magnification factor of ×3, our algorithm accurately recovers the shape of the stripes, while RFDN reconstructs them in completely wrong directions. The experimental results demonstrate that the proposed algorithm can better represent the HR feature space, thereby recovering more high-frequency information in the reconstructed images and bringing them closer to the original HR images.

Furthermore, based on the comparative analysis, it is evident that SISR-RFDM outperforms the other algorithms in terms of the local texture restoration, color saturation, sharpness, and contrast of the reconstructed images. This superior performance can be attributed to the more robust feature representation capability of SISR-RFDM, enabling the extraction of more complex features from the LR space.

### 4.5. Network Parameter Quantity Visualization

To construct a lightweight SR model, the parameters of the network are crucial. We compared our approach with the contrastive algorithms on the test dataset using ×2 magnification as an example and conducted a comparison between the parameter quantity and the PSNR correlation. Additionally, we performed a trade-off analysis between performance and model size, and the results are visualized in Figure 9.

From the figure, it is evident that our algorithm achieves comparable or superior performance while having fewer parameters compared with the other existing techniques. The experimental results demonstrate that SISR-RFDM achieves a better balance between performance and model size.

### 4.6. Comparison with Transformer-Based Algorithms

Compared to CNNs, researchers have attempted to use Transformers to accomplish the task of image super-resolution reconstruction, as demonstrated in LBNet, SwinIR, NGswin, and other models. In this study, our algorithm is compared with these models in terms of parameter quantity and performance metrics, as shown in Table 4, with the test set being ×4 Set5.

In comparison with SwinIR, which has a parameter count of 11.8 M, SISR-RFDM achieves a reduction of 93.31% in parameters while only experiencing a minimal decrease of 0.88% in performance metrics. When compared with the models with fewer parameters, i.e., SwinIR-light and NGswin, SISR-RFDM achieves reductions of 10.23% and 21% in the parameter count, respectively, with corresponding changes in performance metrics of only a 0.03% decrease and a 0.32% improvement. Compared with the model with the fewest parameters, i.e., LBNet, SISR-RFDM sacrifices a small portion of the parameter count to achieve a significant improvement in performance metrics, striking a balance between the parameter count and performance metrics.

## 5. Conclusions

To achieve a better balance between performance and complexity, we propose a lightweight, single-image, super-resolution reconstruction algorithm called SISR-RFDM, which is based on a residual feature distillation mechanism.

The proposed algorithm includes the following key components:

By employing an information distillation structure, the reconstruction of texture in individual images is ensured. This structure aids in extracting and capturing finer and more diverse texture information, thereby enhancing the quality of texture reconstruction in images. Specifically, the distillation layer (DL) and refinement layer (RL) within the information distillation structure allow for a progressive feature extraction process, focusing the model’s learning task more on the reconstruction of image texture details. Additionally, it effectively captures finer and more diverse texture information, enabling the model to better understand the texture information present in the images and improving its ability to reconstruct image texture. Moreover, the exchange of information between the DL and RL layers accelerates the convergence speed of the model and helps to mitigate issues such as gradient vanishing or explosion.

By incorporating the global feature fusion (GFF) structure, our algorithm is capable of enhancing performance while maintaining lightweight characteristics. Specifically, the GFF structure enhances the flow of inter-layer information and promotes feature reuse, resulting in a reduction in network parameters and computational complexity. This is achieved by fusing features from different hierarchical levels, enabling the network to capture information at various scales more effectively while avoiding the excessive computational overhead associated with traditional multi-scale processing approaches. Therefore, with the integration of the GFF structure, our algorithm achieves light weight by extracting image texture details more efficiently while maintaining low computational requirements.

By incorporating a spatial attention (SA) module, it is possible to reduce the number of parameters while retaining crucial spatial information. Specifically, the following enhancements are achieved: Dimension reduction of features: The spatial attention module performs average pooling and standard deviation pooling on the features, extracting the mean and standard deviation of the features, respectively. As these operations are carried out along the channel axis, they lead to a reduction in the feature dimensions. By reducing the feature dimensions, the quantity of model parameters can be significantly decreased, thus achieving light-weighting. Parameter compression: by utilizing a single convolutional layer, features from different positions in the feature descriptor are fused and compressed into a single channel. This compression operation reduces the number of parameters in the model, thereby decreasing the model’s storage requirements and computational complexity and further achieving lightweight. Preservation of spatial information: before feature fusion, the spatial attention module combines the two sets of spatial feature descriptors obtained from average pooling and standard deviation pooling using channel concatenation. This preserves the spatial information of the features, aiding the model in better comprehending and utilizing the spatial structure within the image. As a result, the algorithm maintains high performance while being more efficient and applicable in resource-constrained environments.

Objective quantitative analysis and subjective visual comparisons demonstrate that our proposed algorithm achieves superior results in terms of both subjective visual quality and objective quantification while maintaining relatively low computational complexity compared with the other existing algorithms.

However, this still cannot meet the requirements of practical applications. In future work, we will continue our research in the direction of lightweight models. Additionally, the proposed algorithm only focuses on the super-resolution reconstruction of simple images and does not consider the influence of noise and blur. In future work, we will also explore how to further improve the robustness of the network model in complex application scenarios that involve unknown noise and unknown blur.

## Figures and Tables

**Figure 1 sensors-24-01049-f001:**
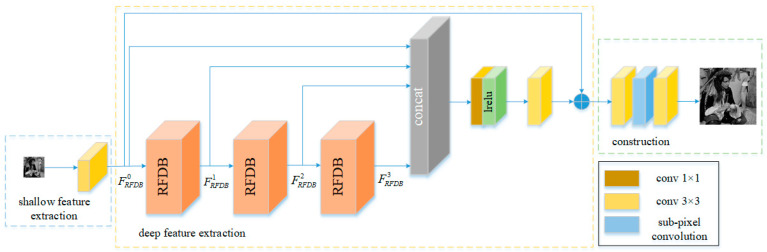
The architecture of a single-image super-resolution network based on the residual feature distillation mechanism.

**Figure 2 sensors-24-01049-f002:**
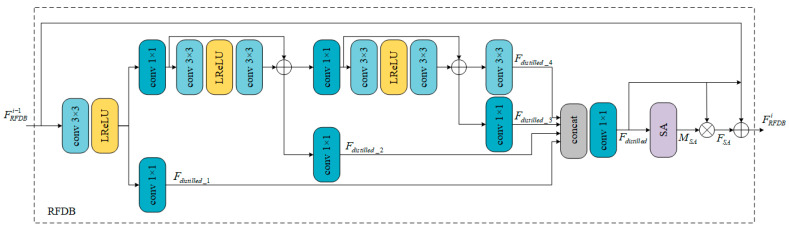
Residual feature distillation block.

**Figure 3 sensors-24-01049-f003:**
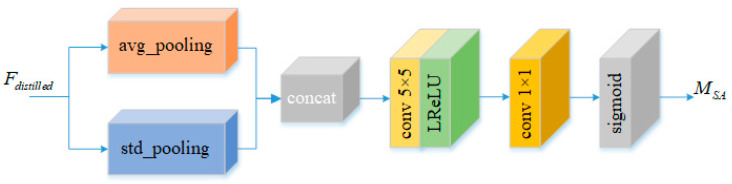
Spatial attention module.

**Figure 4 sensors-24-01049-f004:**
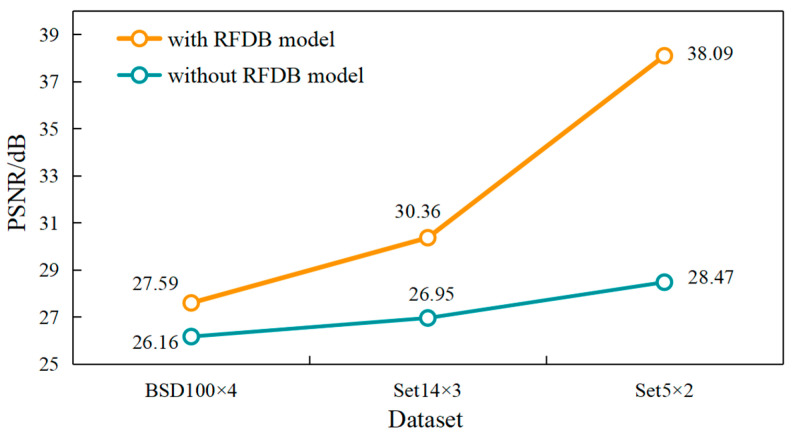
Effect of RFDB module structure on the model.

**Figure 5 sensors-24-01049-f005:**
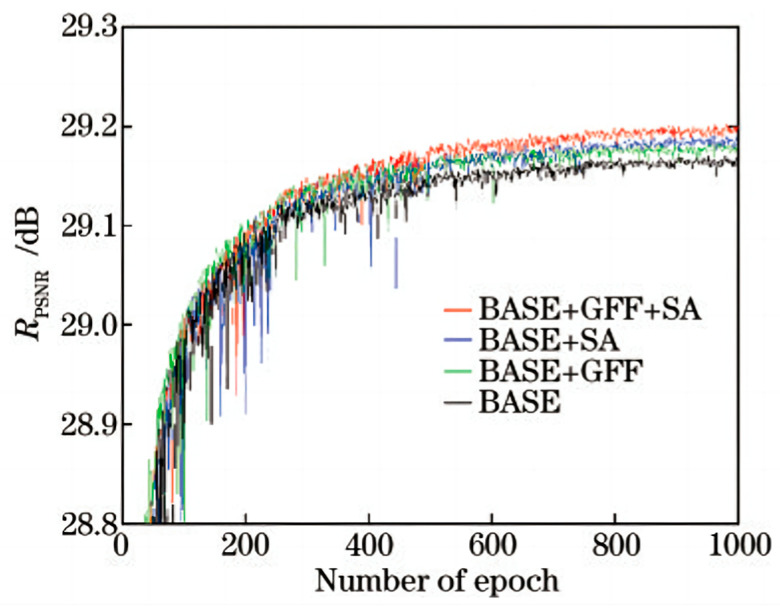
Results of ablation experiments on GFF and SA (validation set).

**Figure 6 sensors-24-01049-f006:**
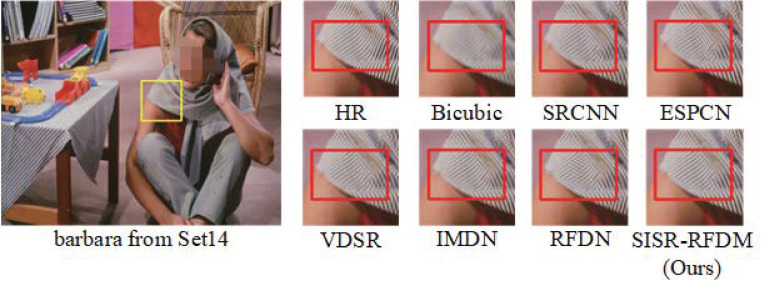
Image visual effects of different algorithms with scale factor ×2.

**Figure 7 sensors-24-01049-f007:**
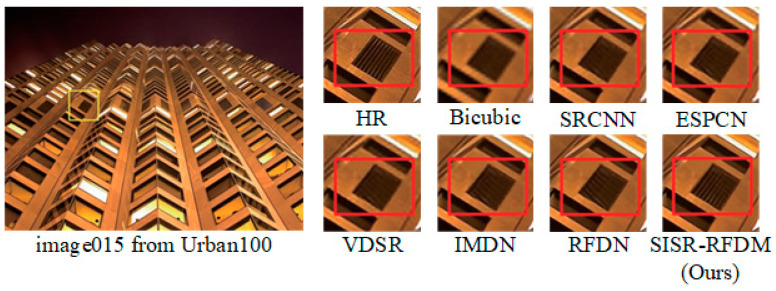
Image visual effects of different algorithms with scale factor ×3.

**Figure 8 sensors-24-01049-f008:**
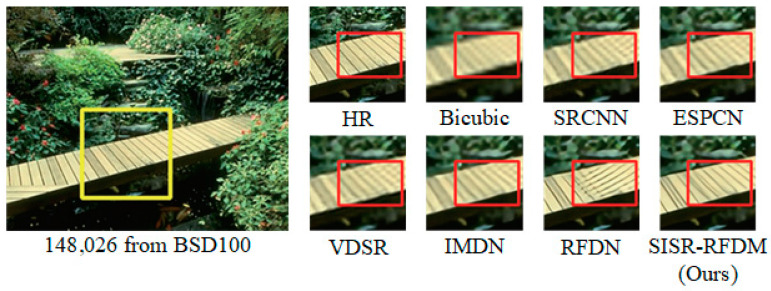
Image visual effects of different algorithms with scale factor ×4.

**Figure 9 sensors-24-01049-f009:**
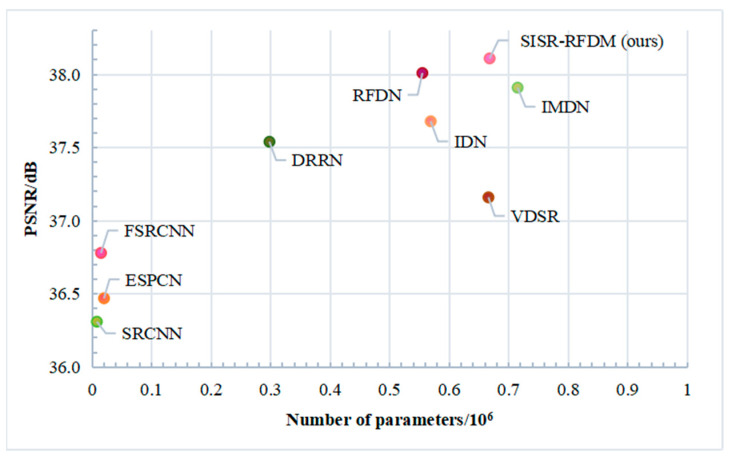
Comparison of network parameters and the PSNR correspondence for different algorithms.

**Table 1 sensors-24-01049-t001:** Results of ablation experiments on GFF and SA (test set).

Scale	Base	GFF	SA	Set5	Set14	BSD100	Urban100
				PSNR/SSIM	PSNR/SSIM	PSNR/SSIM	PSNR/SSIM
×4	√	×	×	32.2029/0.8927	28.6432/0.7826	27.5460/0.7341	26.1329/0.7842
	√	×	√	32.2160/0.8943	28.6571/0.7840	27.5622/0.7357	26.1523/0.7858
	√	√	×	32.2190/0.8946	28.6590/0.7841	27.5631/0.7360	26.1541/0.7861
	√	√	√	**32.2341/0.8958**	**28.6729/0.7843**	**27.5913/0.7362**	**26.1842/0.7864**

Note: Bold is the best result. “√” indicates that the module has been added, and “×” indicates that the module has not been added.

**Table 2 sensors-24-01049-t002:** Average PSNR/SSIM for scale factor ×2, ×3, and ×4 on datasets Set5, Set14, BSD100, and Urban100.

Algotithm	Scale	Set5	Set14	BSD100	Urban100
		PSNR/SSIM	PSNR/SSIM	PSNR/SSIM	PSNR/SSIM
Bicubic	×2	33.69/0.9284	30.34/0.8675	29.57/0.8438	26.88/0.8438
SRCNN [13]	×2	36.31/0.9535	32.26/0.9053	31.13/0.8859	29.30/0.8939
FSRCNN [14]	×2	36.78/0.9561	32.57/0.9089	31.38/0.8894	29.74/0.9009
ESPCN [16]	×2	36.47/0.9544	32.32/0.9067	31.17/0.8867	29.21/0.8924
VDSR [18]	×2	37.16/0.9582	32.87/0.9126	31.75/0.8951	30.74/0.9146
DRRN [20]	×2	37.74/0.9591	33.23/0.9136	32.05/0.8973	31.23/0.9188
IMDN [28]	×2	37.91/0.9594	33.59/0.9169	32.15/0.8987	32.12/0.9278
RFDN [30]	×2	38.05/0.9606	33.68/0.9184	32.25/0.9005	32.19/0.9283
LBNet [33]	×2	-	-	-	-
NGswin [34]	×2	38.05/0.9610	33.79/0.9199	32.27/0.9008	32.53/0.9324
SISR-RFDM (ours)	×2	38.11/0.9613	33.80/0.9193	32.26/0.9006	32.48/0.9317
Bicubic	×3	30.39/0.8682	27.55/0.7742	27.21/0.7385	24.46/0.7349
SRCNN [13]	×3	32.60/0.9088	29. 21/0.8198	28.30/0.7840	26.04/0.7955
FSRCNN [14]	×3	32.51/0.9054	29. 17/0.8181	28.24/0.7821	25.97/0.7917
ESPCN [16]	×3	32.56/0.9073	29. 19/0.8195	28.26/0.7834	25.98/0.7929
VDSR [18]	×3	33.66/0.9213	29.77/0.8314	28.82/0.7976	27.14/0.8279
DRRN [20]	×3	34.03/0.9244	29.96/0.8349	28.95/0.8004	27.53/0.8378
IMDN [28]	×3	34.32/0.9259	30.31/0.8409	29.07/0.8036	28.15/0.8510
RFDN [30]	×3	34.41/0.9273	30.34/0.8420	29.09/0.8050	28.21/0.8525
LBNet [33]	×3	34.47/0.9277	30.38/0.8417	29.13/0.8061	28.42/0.8599
NGswin [34]	×3	34.52/0.9282	30.53/0.8456	29.19/0.8089	28.52/0.8603
SISR-RFDM (ours)	×3	34.55/0.9283	30.54/0.8463	29.20/0.8082	28.66/0.8624
Bicubic	×4	28.42/0.8104	26.00/0.7027	25.96/0.6675	23.14/0.6577
SRCNN [13]	×4	30.22/0.8597	27.40/0.7489	26.78/0.7074	24.29/0.7141
FSRCNN [14]	×4	30.44/0.8595	27.51/0.7507	26.85/0.7090	24.44/0.7188
ESPCN [16]	×4	30.25/0.8566	27.37/0.7487	26.77/0.7072	24.26/0.7114
VDSR [18]	×4	31.35/0.8838	28.01/0.7674	27.29/0.7251	25.18/0.7524
DRRN [20]	×4	31.68/0.8888	28.21/0.7721	27.38/0.7284	25.44/0.7638
SRDenseNet [21]	×4	32.02/0.8934	28.50/0.7782	27.53/0.7337	26.05/0.7819
IMDN [28]	×4	32.21/0.8948	28.57/0.7803	27.54/0.7342	26.03/0.7829
RFDN [30]	×4	32.26/0.8960	28.63/0.7836	27.61/0.7380	26.22/0.7911
LBNet [33]	×4	32.29/0.8960	28.68/0.7832	27.62/0.7382	26.27/0.7906
NGswin [34]	×4	32.33/0.8963	28.78/0.7859	27.66/0.7396	26.45/0.7963
SISR-RFDM (ours)	×4	32.43/0.8972	28.77/0.7858	27.69/0.7406	26.47/0.7980

Note: Red is the best result, while blue is the second-best result.

**Table 3 sensors-24-01049-t003:** Comparison of LPIPS and FID among different algorithms (test set).

Method	Parameters (M)	LPIPS	FID	Time (s)
Bicubic	-	0.602	56.89	0.005
SRCNN [15]	**0.02**	0.444	35.12	0.007
FSRCNN [16]	0.25	0.402	33.92	0.015
ESPCN [18]	0.17	0.376	32.84	**0.004**
VDSR [20]	0.66	0.362	31.92	0.027
DRRN [22]	1.98	0.341	30.72	0.077
IMDN [29]	0.63	0.315	29.67	0.027
RFDN [36]	2.27	0.307	28.89	0.086
LBNet [33]	11.8	0.298	28.41	0.161
NGswin [34]	4.45	0.297	28.38	0.049
SwinIR-light [47]	1.52	0.292	28.15	0.016
SISR-RFDM (ours)	0.77	**0.281**	**27.38**	0.017

Note: Bold is the best result.

**Table 4 sensors-24-01049-t004:** Comparison with Transformer-based algorithms.

Model	Parameters (M)	PSNR (dB)	SSIM
LBNet [33]	**0.72**	32.29	0.8960
SwinIR [47]	11.80	**32.72**	**0.9021**
SwinIR-light [47]	0.88	32.44	0.8976
NGswin [34]	1.00	32.33	0.8963
SISR-RFDM (ours)	0.79	32.43	0.8972

Note: Bold is the best result.

## Data Availability

The original contributions presented in the study are included in this article. Further inquiries can be directed to the corresponding author.

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
