# Peer review of "A Lightweight Image Super-Resolution Reconstruction Algorithm Based on the Residual Feature Distillation Mechanism"

_sensors, 2024, doi:10.3390/s24041049_

Round 1
Reviewer 1 Report
Comments and Suggestions for Authors
The author proposed a lightweight single-image super-resolution reconstruction algorithm called SISR-RFDM, based on a residual feature distillation mechanism, to achieve a better balance between performance and complexity. My detailed comments are listed below, which hopefully can help the authors improve the quality of their work.
(1) In the introduction part, some references are relatively old. Some related works should be further systematically surveyed.
(2) I suggest that the author describe in more detail which part of the network does Global feature fusion (GFF) correspond to? It is the deep feature extraction module in Figure 1, or the concat operation on the features extracted by RFDB in that module?
(3) In the ablation study part, it is suggested to add the ablation experiment on the residual feature distillation block(RFDB) to verify its effectiveness.
(4) Please check the formula writing carefully. For example, in Section 3.2.1, formula (6) (7) is inconsistent with the flow shown in Figure 2, and formula (8) (9) is repeated.
Comments on the Quality of English LanguageMinor editing of English language required.
Author Response
Please see the attached file: Response to Reviewer 1.pdf.

Reviewer 2 Report
Comments and Suggestions for Authors
In this paper, the authors propose a super-resolution reconstruction algorithm named SISR-RFDM for image super-resolution. It is improved based on the residual feature distillation mechanism (RFDM) and by introducing spatial attention (SA) modules to recovering high-frequency details. Global feature fusion (GFF) then is applied to enhance inter-layer information flow and feature reuse for the output of each residual block.
The paper focuses on CNN-based super-resolution algorithms, which have some novelty. After carefully reading the entire manuscript, I have the following questions:
(1) The title of the manuscript includes "lightweight" model, but the relevant experiments and methods have not fully reflected the characteristics of “lightweight”.
(2) The super-resolution algorithms compared in the manuscript all appear before 2021. It somewhat affects the innovation in the manuscript. It would be more convincing to add comparisons with some SOTA lightweight models such as super-resolution algorithms based on GAN, Transformer, Diffusion, and other models.
(3) In the evaluation indicators, the author only listed PSNR and SSIM. In fact, there are other indicators such as LPIPS, FID, etc. If more comparison of evaluation indicators could be made, it would be more convincing.
Author Response
Please see the attached file: Response to Reviewer 2.pdf.

Reviewer 3 Report
Comments and Suggestions for Authors
In this paper, the author proposed a lightweight image super-resolution reconstruction method based on residual feature distillation mechanism. Experiments show that the peak signal-to-noise ratio is improved by 0.23dB, and the structural similarity index reaches 0.9607. The paper is well thought out and worked on. But I also have some suggestions.
1) For the innovation of the proposed image super-resolution reconstruction algorithm, it seems that the description is not sufficient. It should be described in details for better understanding.
2) The texture restoration is significant for image super-resolution reconstruction. Please give more descriptions on how to ensure the texture reconstruction of the single images by the proposed method.
3) The author only provides three visual comparison results, which cannot fully demonstrate the effectiveness of the proposed algorithm.
4) Why use residual feature distillation mechanism? Please provide a more detailed description.
5) How to determine parameters β1 and β2?
6) Can this method be extended to stereoscopic image super-resolution?
7) The size of all formulas is too large.
8) Reviewing throughout the paper, the English grammar, spelling, and sentence structure still needs improve.
Comments on the Quality of English LanguageMinor editing of English language required.
Author Response
Please see the attached file: Response to Reviewer 3.pdf.

Round 2
Reviewer 1 Report
Comments and Suggestions for Authors In this manuscript, all my concerns have been addressed and I have no further questions.Reviewer 2 Report
Comments and Suggestions for Authors
In this version of the manuscript, the author has provided detailed answers to the content I have been concerned about. I have no further questions and think it can be published.
Reviewer 3 Report
Comments and Suggestions for Authors The authors have addressed my concerns, so I suggest accepting it as a regular paper.